Pepper power: short-term impact of pepper consumption on the gut bacteriome composition in healthy volunteers

http://orcid.org/0009-0007-8470-4708 Owolo Oluwafayoke 1
Audu Haruna J. 1
http://orcid.org/0000-0003-1405-2365 Afolayan Ayorinde O. 2
http://orcid.org/0000-0002-2379-0135 Ayeni Funmilola A. 3 fayeni@iu.edu
1 Department of Pharmaceutical Microbiology, Faculty of Pharmacy, University of Ibadan , Ibadan, Oyo State , Nigeria
2 Institute for Infection Prevention and Control, Medical Centre, University of Freiburg , Freiburg im Breisgau , Germany
3 Environmental and Occupational Health, School of Public Health, Indiana University , Bloomington, IN , United States
Nakai Kenta
Electronic publication date: 2024 Dec 13
Publication date: 2024
Volume: 12
Electronic Location ID: e18707
Received 2024 Sep 11; Accepted 2024 Nov 23
Copyright: © 2024 Owolo et al.
Copyright year: 2024
Copyright holder: Owolo et al.
License: This is an open access article distributed under the terms of the Creative Commons Attribution License, which permits unrestricted use, distribution, reproduction and adaptation in any medium and for any purpose provided that it is properly attributed. For attribution, the original author(s), title, publication source (PeerJ) and either DOI or URL of the article must be cited.
License URL: https://creativecommons.org/licenses/by/4.0/

Keywords: Pepper, 16S sequencing, Capsicum chinense, Gut microbiota

Funding: Vassar College, Poughkeepsie, NY, USA Indiana University Bloomington, IN, USA Federal Ministry of Education and Research (BMBF) 01KI2018 An internal grant from Vassar College, Poughkeepsie, NY, USA and Indiana University Bloomington, IN, USA was used for some aspects of this study. Ayorinde O. Afolayan is a post-doctoral researcher supported by the Federal Ministry of Education and Research (BMBF) under 01KI2018. The funders had no role in study design, data collection and analysis, decision to publish, or preparation of the manuscript.

==============================
Background

Pepper from Capsicum species is a well-established spice with a rich history of culinary use. Some observations have linked its consumption to gastrointestinal discomfort and alterations in stool patterns while it is considered beneficial in some cultures. However, there is lack of information on the direct effect of pepper consumption on human gut microbiota, we conducted dietary intervention studies to assess the impact of pepper on gut bacteriome composition in humans.

Methods

Ten healthy volunteers were recruited, and each person received 200 ml of 0.14 g/ml fresh Habanero Pepper (Capsicum chinense) daily over a 4-day period after which they abstained from pepper consumption for the subsequent 4 days before resumption of their normal diet. Stool samples were collected at baseline, after pepper consumption, after 4 days without pepper and after 4- and 6-days resumption of normal diet. We sequenced the V3-V4 region of the 16S rRNA gene and analyzed microbial diversity and composition using the QIIME2 pipeline and relevant R packages.

Results

Consumption of pepper over a 4-day period led to a higher abundance of Verrucomicrobia, a phylum rarely found in significant proportions at other time points. There was a gradual depletion of Shigella and Staphylococcus spp. from baseline untill the end of the study. Other taxa showed timepoint specific associations, emphasizing the potential impact of short-term dietary interventions on the relative abundance of these genera.

Conclusions

Our study adds nuance to the understanding of diet-microbiota interactions, highlighting the intricate relationship between pepper consumption and gut bacteriome composition. Further exploration of these dynamics holds promise for personalized dietary recommendations and targeted interventions to support gut microbial health.

Introduction

The human gut microbiota, composed of diverse microorganisms inhabiting the gastrointestinal tract, plays a pivotal role in maintaining health and preventing disease (Ayeni et al., 2018). This complex ecosystem, influenced by a myriad of factors, including genetics, age, and environmental exposures, has been the subject of extensive research due to its profound implications for human well-being (Cénit et al., 2014; Cahana & Iraqi, 2020). Among these factors, diet stands out as a significant modulator of gut microbial composition and function (Zmora, Suez & Elinav, 2018; Kurilshikov et al., 2021). Dietary choices not only shape the diversity of microbial species residing in the gut but also influence their metabolic activities, impacting various aspects of host physiology, including immune function, metabolism, and even mental health (David et al., 2014; Berding et al., 2021). Thus, understanding the intricate interplay between diet and the gut microbiota has become a paramount area of investigation in contemporary biomedical research.

One food item that has garnered attention for both its culinary appeal and potential health implications is pepper, derived from various species of the Capsicum genus. The Capsicum genus comprises more than 200 varieties; the five major species are C. annuum, C. baccatum, C. chinense, C. frutescens and C. pubescens., with C. chinense reported by some authors as the hottest pepper in the world (Sosa-Moguel et al., 2017). The active compound in pepper is capsaicin and it is involved in its health properties (Rosca et al., 2020). Peppers belonging to the genus Capsicum have characteristic pungency and flavor with a long history of human use. They are recognized as one of the earliest spices employed in cooking (Perry et al., 2007; Hill et al., 2013). Pepper is a ubiquitous spice that is used to enhance the flavors of numerous regional and global dishes. Beyond its culinary significance, pepper has been recognized for its potential health benefits, echoing ancient beliefs in spices as remedies for humoral imbalances, a concept rooted in the teachings of Hippocrates and Galen (Nam & Nam, 2014).

Modern scientific investigations, both in vitro and in vivo, have substantiated the therapeutic potential of Capsicum sp. its bioactive constituents, and phytochemicals (Ao, Huang & Liu, 2022). The phytochemical activity of pepper compounds, such as capsaicin and piperine, has been associated with various health-promoting properties, including anti-inflammatory, antioxidant, and antimicrobial effects (Dludla et al., 2023). Its antimicrobial properties have been demonstrated against pathogens such as Helicobacter pylori, other bacteria, and fungi (Lai & Roy, 2012; Tharmalingam et al., 2014; Füchtbauer et al., 2021; Manjunath et al., 2022). Additionally, emerging research suggests that Capsicum sp. pepper may exert profound effects on the intestinal microbiota and influence the pathogenesis of metabolic disorders (Qin et al., 2012; Tang et al., 2013; Rosca et al., 2020).

However, despite its popularity, the consumption of Capsicum sp. pepper has been occasionally associated with gastrointestinal discomfort, alterations in stool patterns, and could ultimately result in gastrointestinal disease conditions (Esmaillzadeh et al., 2013; Chaiyasit & Wiwanitkit, 2016). This momentary diarrhea has been attributed to the irritation of the digestive system (stomach and intestinal linings) by capsaicin, a potent ingredient in chili peppers which triggers certain receptors in the digestive system, resulting in burning sensation followed by rapid and hence inadequate processing of the food (Chaiyasit & Wiwanitkit, 2016; Xiang et al., 2022). Conversely, other studies have reported the alleviating effect of Capsicum sp. pepper in calves, pigs, and piglets (Cairo et al., 2018; Moraes et al., 2022; Su et al., 2023; Satitsri et al., 2023). However, there is lack of information about the direct effect of pepper consumption in human. This lack of information has prompted scientific inquiry into the effects of Capsicum sp. pepper consumption on gut health and its influence on the composition of the human intestinal microbiota.

This study seeks to address the potential effects of short-term dietary intervention with pepper on the gut bacteriome composition in humans. We aimed to elucidate whether the consumption of pepper leads to discernible alterations in the microbial landscape of the human gut. Additionally, we sought to identify any associations between pepper intake and the abundance of specific bacterial l taxa.

Materials and Methods

Pepper species used in this study

We used C. chinense in this study due to a previous report of its being a very hot pepper genus (Sosa-Moguel et al., 2017).

Ethics statement

The Oyo State Ministry of Health granted ethical approval for the human study (Reference Number: AD 33/478/654; Approval date: 19th February 2019).

Recruitment of volunteers

A total of 10 individuals (five males, five females) were recruited into this study using convenience sampling. Verbal and written informed consents were obtained from the volunteers before the study. Subjects were healthy postgraduate students and staff (aged 18–49) from the department of Pharmaceutical Microbiology, Faculty of Pharmacy, University of Ibadan, Ibadan, Oyo state, Nigeria. The volunteers were healthy, do not consume alcohol, and had not been treated for any disease using antibiotics for at least 1 month before sample collection. The participants normally consume moderate amount of pepper habitually and they continued with their normal diets throughout the study. The pepper dose chosen for the study was higher than their normal daily consumption. We used 200 ml of 0.14 g/ml of C. chinense pepper which was milled, cooked and consumed as soup one time daily, during the afternoons of each day. We used this amount after interacting with the volunteers about the quantity of pepper that is higher than their daily intake but that they can tolerate without adverse effects.

Sample collection

Individuals recruited for the study consumed 200 ml of 0.14 g/ml of fresh Habanero Pepper (Capsicum chinense) once each day for the first 4 days after which they abstained from all form of pepper consumption for the subsequent 4 days. The last stage involved the subjects resuming their normal pepper consumption in their diet, 4 and 10 days into which samples were likewise collected (Fig. 1). Fecal samples were obtained from all 10 volunteers at five time points (initial, after 4 days of pepper consumption, after 4 days without pepper, after 4 days of a normal diet, and after 10 days of a normal diet). The fecal samples were preserved according to the method reported by Ayeni et al. (2018) and subsequently transported to Vassar College, Poughkeepsie, NY, USA for further analysis.

Figure 1 Timepoints in the study.

16S rRNA gene sequencing and analysis

DNA was extracted from fecal samples collected during the intervention study using the QIAamp DNA Stool Mini Kit (QIAGEN, Hilden, Germany) following the manufacturer’s protocol. Amplification of the V3–V4 region of the 16S rRNA gene was done using the 341F and 805R primers with Illumina adaptor overhang sequences as previously described (Candela et al., 2016). Library was prepared and sequencing was performed on Illumina MiSeq platform using a 2 × 300 bp paired-end protocol, according to the manufacturer’s instructions (Candela et al., 2016).

Sequence data were imported into QIIME2 (https://qiime2.org; version 2020.8.0) (Bolyen et al., 2018) and processed. The DADA2 Plugin was used for quality control and chimera removal. DADA2 was used to trim low-quality reads with a sampling depth of 1,000. Taxonomy assignment was achieved using the q2-feature-classifier plugin—a pre-trained naïve Bayes classifier, and the SILVA 138 reference database (Pruesse et al., 2007) for 99% of the V3–V4 16S rRNA gene region. QIIME2 artifacts were exported into RStudio version 2022.02.3 (R Core Team, 2022, R version 4.2.2) and transformed into a Phyloseq object using the Phyloseq R package, version 1.38.0 (McMurdie & Holmes, 2013) for further downstream analyses, statistics, and visualizations. Normalization was achieved using the transform_sample_counts function in Phyloseq. Alpha diversity indices was calculated using the Phyloseq R package according to the following metrics: Pielou’s evenness metrics (Pielou, 1966), Chao1 (Chao, 1984), and Shannon’s index (Shannon & Weaver, 1949). Beta diversity was determined by calculating Weighted and Unweighted Unifrac distances (Lozupone & Knight, 2005), as well as Jaccard (Jaccard, 1908) and Bray Curtis (Bray & Curtis, 1957) distances. Beta diversity was visualized by Principal Co-ordinate Analysis using the Phyloseq R package. The QIIME2 adonis plug in was used to validate the calculation of the Weighted and Unweighted Unifrac distances, Jaccard similarity index and Bray Curtis matrix as measures of Beta diversity.

Statistical analyses

Statistical analysis was performed using GraphPad Prism version 8.0.2 for Windows (GraphPad Software, La Jolla, CA, USA, www.graphpad.com). Normality of data was assessed (Ghasemi & Zahediasl, 2012) using the Shapiro test function from the stats package in R (v4.2.2) (R Core Team, 2022). In all the phases, Pearson correlation test was carried out. Pairwise PERMANOVA comparisons were carried out to estimate microbial community diversity in the context of categorical metadata (pepper interventions), using the q2-diversity plugin of QIIME2. The Wilcoxon signed-rank test was applied to compare taxonomic abundance at the phylum and genus levels among each pair of intervention groups. This comparison was conducted in R (v4.2.2; R Core Team, 2022) using several functions: stat_compare_means from the R package ggpubr (v0.4.0) (Kassambara, 2023a), wilcox_test from rstatix (v0.7.2) (Kassambara, 2023b), and pairwise wilcox test from stats package. Additionally, the Friedman test was employed to evaluate alpha diversity and taxonomic abundance across all intervention groups, utilizing the Friedman test function from the stats package in R. All statistical comparisons were corrected for multiple testing using the Holm’s method, available in the rstatix package and stats package.

Results

Ten healthy adults (five males and five females, mean age = 27.3 years) provided 52 fecal samples in total. The subjects underwent four pepper consumption phases: baseline, higher pepper consumption, no pepper consumption, and normal pepper consumption. The V3–V4 of the microbial 16S rRNA gene was sequenced using the Illumina MiSeq platform, generating a total of 651,463 high-quality paired-end reads, with an average of 12,528 reads per subject. These reads were further clustered into 1,838 OTUs/ASVs.

We confirmed that the data significantly deviated from the normal distribution (p = 1.089432 × 10−91), as is expected for microbiome data. Alpha diversity indices, including Chao1, Pielou’s evenness, and Shannon entropy index, displayed no significant variation across all time points (p > 0.05, all groups, and pairwise). Additionally, UniFrac (Weighted and Unweighted), Bray Curtis, and Jaccard emperor did not reveal distinct clusters between different intervention time points.

The most abundant genera present in all groups were Corynebacterium (17.7–41.2%, 25.6 ± 3.5), Bacteroides (7.7–36.45%, 24.1 ± 3.9), and Mycoplasmas (10.71–28.67%, 16.4 ± 2.7) (Fig. 2). During the 10-day normal pepper consumption there is heightened Bacteroides populations (36.45%). Shigella and Staphylococcus, initially present (12.43% and 6.6%), decreased in subsequent interventions, with the lowest recorded abundance in the 10-day normal pepper consumption time point (3.79% and 1.6% respectively) (Fig. 2). Streptococcus, Weissella, and Acinetobacter presented in minimal levels solely in the gut of volunteers during the 4-day pepper consumption phase, 4-day normal pepper consumption phase, and 10-day normal pepper consumption phase, respectively (Table S1).

Figure 2 Relative abundance of most abundant genera at different time points.

Members of Bifidobacteriaceae showed a potential differential abundance during the period of abstaining from pepper for 4 days compared to the time of returning to the normal pepper consumption for 4 days (p = 0.005) and 10 days (p = 0.006). Similarly, members of Bacteroidaceae (p = 0.038) displayed a differential potential during the period of consuming pepper for 4 days compared to those reverting to the normal pepper consumption for 4 days (p = 0.015), albeit not significant upon adjusting the p-values (Table S2).

At the phylum level, the microbiota composition across all time points showed notable abundance of Firmicutes (19.6 ± 19.6), Bacteroidetes (27.2 ± 26), Proteobacteria (10.5 ± 6.5), Actinobacteria (18.3 ± 10.5), and Tenericutes (21.2 ± 13). Notably, participantshad increased levels of Firmicutes and Bacteroidetes on returning a 10-day normal pepper consumption. During the pepper intervention stage, Verrucomicrobia dominated, constituting 5.6% of the microbial composition (Fig. 3, Table S3).

Figure 3 Relative abundance of bacterial phylum at different time points.

Significant variations were observed in the abundance of Tenericutes (p = 0.024), which appeared notably lower during the baseline period (Initial) compared to the period when pepper was consumption was increased for 4 days. Additionally, Firmicutes showed a lower abundance (p = 0.000788) during the baseline period compared to the samples at the 4-day normal diet, while Proteobacteria exhibited lower abundance (p = 0.013) during the baseline period compared to the samples at the 10-day normal pepper consumption. However, upon correction for multiple testing, no significant differences in the abundance of these phyla were found between the study time points (Table S3).

Discussion

We explored the influence of short-term dietary interventions on the diversity and abundance of the gut bacteriome in a diverse group of healthy individuals. The gut microbiome’s dynamic nature and its role in human health make it a compelling target for investigations into the impact of diet. This aligns with a previous study (David et al., 2014) highlighting that dietary changes can promptly alter gut microbiota composition, evident within 1‒3 days of intervention. The study further established that even short-term dietary interventions consisting of plant and animal products can reshape microbial communities and override inter-individual differences in microbial gene expression. Although many dietary intervention studies extend over 15–30 days, our study, in concordance with prior research, underscores that diet modifications can swiftly alter gut microbiota composition and population within a few days. Our analysis of alpha and beta diversity metrics provided valuable insights into the response of gut microbial communities to different dietary regimens. We did not observe significant variations in alpha diversity metrics, including Chao1, Pielou’s evenness, and Shannon entropy index, across all dietary times. These results suggest that the overall richness and evenness of microbial communities remained relatively stable, regardless of short-term dietary changes. This finding aligns with previous studies that highlight the robustness of the gut microbiota in response to acute dietary alterations (Lozupone et al., 2012; Sommer et al., 2017; Fragiadakis et al., 2020; Fassarella et al., 2021; Jian et al., 2021).

However, an intriguing observation emerged concerning the influence of pepper consumption on the gut microbiota. The gradual depletion of Shigella and Staphylococcus from baseline is an interesting observation. A possible reason could be the antimicrobial effects of the pepper, which has a lingering effect till 10 days of resumption of normal diet, testifying to its antimicrobial effects on some specific pathogenic microbes (Füchtbauer et al., 2021; Manjunath et al., 2022). However, this warrants further investigation. Consumption of higher amount of pepper over a 4-day period exhibited a higher abundance of Verrucomicrobia, a phylum rarely found in significant proportions in other dietary time. Phylum Verrucomicrobiota are Gram-negative bacteria that have been isolated from fresh water, soil environments and human faeces. Lower proportions of phylum Verrucomicrobia has been associated with poorer sleep quality in the older people and in prediabetes (Barlow, Yu & Mathur, 2015; Anderson et al., 2017). Interestingly, the beneficial Akkermansia muciniphila which has been noted for its beneficial activities like anti-inflammatory and immunostimulatory properties, improvement of gut barrier function and insulin sensitivity is a member of phylum Verrucomicrobia (Cani et al., 2022). The dominance of this beneficial phylum in this study during pepper consumption requires further investigation into potential prebiotic and beneficial properties of pepper.

At the genus level, Bacteroides, Corynebacterium, and Mycoplasmas were consistently abundant across all time points. These genera appeared to be resilient to short-term dietary changes, this suggests their potential roles as core members of the gut ecosystem. In contrast, certain genera, such as Streptococcus, Weissella, and Acinetobacter, exhibited time point-specific associations, indicating that short-term dietary interventions could influence the relative abundance of certain microbial taxa. Notably, members of the Enterobacteriaceae family were not prevalent and were not significantly abundant across the dietary groups, a phenomenon that could be linked to the dominance of the other genera. The persistent abundance of these genera might create a competitive environment, potentially restraining the proliferation of Enterobacteriaceae. It’s important to note that the dynamics of gut microbial communities are intricate, and there is still a lot to be understood about interplay between different microbial species, as well as their responses to external factors like dietary components.

Importantly, our study has some limitations that warrant consideration. The relatively small sample size and the short duration of dietary interventions may have limited our ability to capture more subtle changes in microbial diversity and composition. Long-term dietary interventions and larger cohorts are needed to further investigate the complex and dynamic interactions between diet and the gut microbiome. Despite these limitations, our pilot study contributes valuable information to the growing body of research on diet-microbiome interactions. Understanding the impact of dietary interventions on the gut microbiome has implications for personalized dietary recommendations and the development of targeted interventions to promote gut microbial health. Future studies with more extended intervention periods and diverse dietary profiles will be essential for unraveling the intricate relationships between diet and the gut microbiota and their potential impact on human health.

Conclusions

Our study contributes valuable insights to the field of diet-microbiota interactions, with implications for personalized dietary recommendations and the development of targeted interventions aimed at supporting gut microbial health. We provided baseline data for dietary intervention in humans with Capsicum sp. pepper consumption, which has been lacking. The findings presented herein provide a foundation for further research in this burgeoning field, offering the potential for novel strategies to promote human well-being through dietary modulation of the gut microbiota.

Supplemental Information

Supplemental Information 1 Abundance of microbial genus at different time points.

Supplemental Information 2 Abundance of microbial family at different time points.

Supplemental Information 3 Abundance of microbial phyla at different time points.

Additional Information and Declarations

Competing Interests

Author Contributions

Human Ethics

Data Availability

The authors declare that they have no competing interests.

Oluwafayoke Owolo performed the experiments, authored or reviewed drafts of the article, and approved the final draft.

Haruna J. Audu performed the experiments, analyzed the data, prepared figures and/or tables, authored or reviewed drafts of the article, and approved the final draft.

Ayorinde O. Afolayan performed the experiments, analyzed the data, prepared figures and/or tables, authored or reviewed drafts of the article, and approved the final draft.

Funmilola A. Ayeni conceived and designed the experiments, authored or reviewed drafts of the article, and approved the final draft.

The following information was supplied relating to ethical approvals (i.e., approving body and any reference numbers):

Oyo State Ministry of Health granted ethical approval for the study (Reference Number: AD 33/478/654; Approval date: 19th February 2019).

The following information was supplied regarding data availability:

The raw sequence data is available at NCBI: PRJNA1054999.

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
