# Peer review of "Pepper power: short-term impact of pepper consumption on the gut bacteriome composition in healthy volunteers"

_PeerJ, doi:10.7717/peerj.18707_

## Round 0.1 · original submission · Major Revisions

Two experts in the field reviewed your manuscript. As you can see from their comments below, both proposed many points to improve, though they are not fundamental. Please read their comments carefully and revise the manuscript accordingly. I also felt that you should record more details of the pepper used and how they were consumed.

I am looking forward to your revised manuscript.

Reviewer 1 ·

Basic reporting

-There are some grammatical issues, for example:
line 37: Edit to "...microbiota. We conducted..."
line 38: Edit to "...in humans."
line 41: I would edit to "...abstained from pepper consumption..." to clarify the meaning of the sentence.
line 47: "... at other time points."
line 48: "...from baseline until the end of..."
line 113: "... in humans."


-Line 142: The consumption of the "milled and boiled" peppers is a little confusing to me as it's stated. Was it a "mash" that the participants drank? Were they given any water or anything before or after consumption? More details about this would be good.

-Line 142: Please clarify why you boiled the peppers.

-Line 155: Change from "16S rRNA" to "16S rDNA".

-Line 158: Why was the V3-V4 region used?

-Line 171: Change "Alpha diversity index was..." to "Alpha diversity indices were..."

-In the "Statistical Analyses" section, please add citations for the R packages that were used.

-Line 209: Consider making relative abundance box plots for the highly represented genera so that this data can be assessed visually by the reader.

Experimental design

-Consider adding some context to how much of the pepper microbiome may have made it to the stool samples. For example, Pepper Mild Mottle Virus is used to demonstrate that human fecal matter is present in some studies. This suggests that even cooked vegetables may have remaining microbial DNA. It would benefit the paper to discuss this.

Validity of the findings

-Line 260: Since the authors are drawing particular attention to OTUs that were assigned to Shigella and Staphylococcus, it would be good to include figures that show those genera in a figure in the main text in addition to the table in the supplement.

Additional comments

- Line 79 and throughout the work - I apologize if this is clearer to others, but I'm a little confused about what the authors mean when they say pepper. Do you mean black pepper flakes/grinds? Or do you mean pepper oils derived from the pepper flesh? Do you mean pepper flesh? If you are just talking about the general consumption of peppers, I would use the plural of the word. You might also give examples of how peppers are typically consumed to help clarify what you mean for readers.

-You specify the pepper species you had study participants eat. Did you ask the participants about all the species of pepper they typically consume? Are there other chemical compounds in different species of pepper that might change the impacts observed?

-Lines 264-266: This sentence is unclear and worded like you are comparing eating peppers over 4 days to a different experimental treatment group, whereas you are discussing a timepoint among all participants. This happens a few times in the text, consider rewording.

·

Basic reporting

Overall, the article is well-written; however, in some sections (for instance, lines 264 to 274), the phrasing becomes somewhat unclear. I recommend an English language review before resubmission.

Experimental design

the authors consider alternating the two interventions (i.e., with and without chili pepper) with a washout period (a normal diet) in order to better distinguish the effects of each diet in comparison to the baseline.
In the sample collection section (line 146), there is no mention of the conditions under which samples were maintained from collection to arrival at the lab. Were samples stored at 4°C or room temperature? This information is essential.

Validity of the findings

In line 165, the authors mention using the DADA2 pipeline with a depth of 1000. I am uncertain of the intended reference here. Does this number represent the average number of reads after DADA2 processing, or was it chosen specifically for beta-diversity analysis? In either case, it seems a relatively low value for such analyses. Additionally, I could not find any mention in the manuscript of data normalization prior to statistical analysis, which is a critical step for accurately make any consideration about samples diversity.

---

## Round 0.2 · Minor Revisions

Although I could not get the new comments from the original Reviewer 1, I confirmed that the revisions were done appropriately. Although Reviewer 2 has been almost satisfied with the revisions, he/she pointed out one minor point. Please correct the MS if you think this comment makes sense.

·

Basic reporting

In line 357, the manuscript refers to Figure 2 as 'Relative abundance of phylum-level taxa. Bars are faceted by the dietary phase timeline.' However, this should now refer to Figure 3 due to changes in the manuscript. Please update the figure title accordingly and provide a title for the new Figure 2, 'Top genera.'

Experimental design

no comment

Validity of the findings

no comment

Additional comments

no comment

---

## Round 0.3 · accepted · Accept

I confirmed your changes and will recommend the acceptance of your MS. Congratulations!